# *Does Incomplete Syntax Influence Korean Language Model?* Focusing on Word Order and Case Markers

**Jong Myoung Kim**[1]    **Young-Jun Lee**[1]

**Yong-jin Han**[2]    **Sangkeun Jung**[3]    **Ho-Jin Choi**[1] *

[1]School of Computing, KAIST
[2]School of Computer Science and Engineering, Kyungpook National Univeristy
[3]The Division of Computer Convergence, Chugnam National University

grayapple@kaist.ac.kr    yjhan@sejong.knu.ac.kr

{yj2961,hojinc}@kaist.ac.kr    hugman@cnu.ac.kr

## Abstract

Syntactic elements, such as word order and case markers, are fundamental in natural language processing. Recent studies show that syntactic information boosts language model performance and offers clues for people to understand their learning mechanisms. Unlike languages with a fixed word order such as English, Korean allows for varied word sequences, despite its canonical structure, due to case markers that indicate the functions of sentence components. This study explores whether Korean language models can accurately capture this flexibility. We note that incomplete word orders and omitted case markers frequently appear in ordinary Korean communication. To investigate this further, we introduce the Syntactically Incomplete Korean (SIKO) dataset. Through SIKO, we assessed Korean language models' flexibility with incomplete syntax and confirmed the dataset's training value. Results indicate these models reflect Korean's inherent flexibility, accurately handling incomplete inputs. Moreover, fine-tuning with SIKO enhances the ability to handle common incomplete Korean syntactic forms. The dataset's simple construction process, coupled with significant performance enhancements, solidifies its standing as an effective data augmentation technique. We make our source code and dataset publicly available. [1]

## 1 Introduction

Syntactic information is crucial for humans to understand sentence meanings accurately (Chomsky, 2002). One prominent syntactic element, *word order*, has been a central topic in recent Language Model (LM) research. Studies have explored various aspects of word order: its significance to LMs (Pham et al., 2020; Sinha et al., 2020), methods to improve model performance via word order adjustments (Sinha et al., 2021), and insights into how LMs progressively learn human language (Abdou et al., 2022).

Korean follows a canonical Subject-Object-Verb (S-O-V) word order. However, the presence of case markers, a type of postposition, allows Korean to exhibit relatively flexible word order (Lee & Im, 1997; Yeon & Brown, 2013; Sohn, 2005). Case markers denote the grammatical roles of constituents within a sentence, like their function as a subject or object. Conversely, due to its canonical word order, Korean often experiences omissions of case markers. Our aim is to verify whether this flexibility is reflected in Korean LMs.

---

*Corresponding author.
[1]https://github.com/grayapple-git/SIKO

Figuree 1 exemplifies the flexibility of a Korean sentence, "나비가 꿀을 마신다," in terms of word order and case markers. Even with word order changes (Arrow a), the case markers '가' and '을' identify '나비' and '꿀' as subject and object, respectively. Conversely, in S-O-V order (Arrow b), subject and object are identifiable by position, even without case markers. Of course, the freedom from case markers and word order comes with limitations. Sentences lacking case markers or with rearranged word order might appear unnatural or have the potential to be interpreted differently from the original sentence. More detailed information about Korean word order and structure is as follows:

- Korean typically follows an SOV (Subject-Object-Verb) order.
- Major reorderings occur among full phrases (e.g., noun phrase arguments) and usually take place before the verb.
- Moving the verb to an earlier position is much rarer.
- Splitting a noun phrase argument (e.g., placing an adjectival modifier before the noun) is also much rarer.
- Naturally dropping case markers is more common with subjects than with objects (or vice versa, depending on the context).

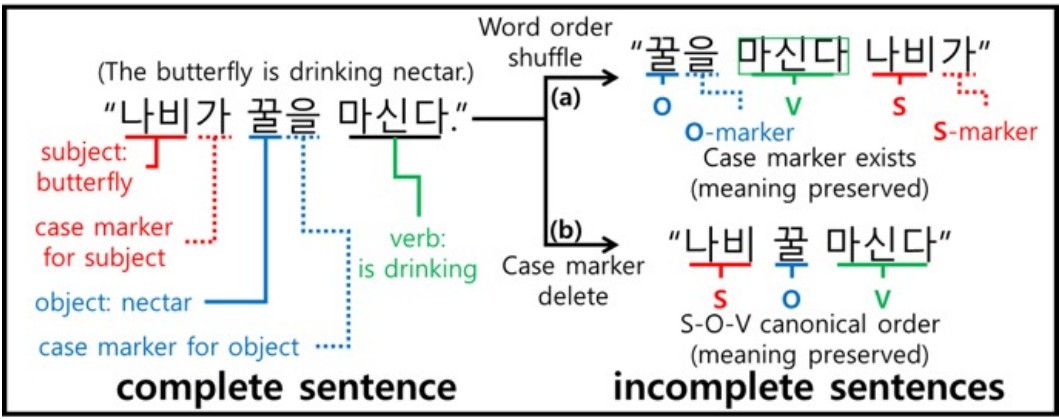

Figure 1: An example of Korean syntax flexibility: a) Case markers enable word order variability, b) They can be omitted in canonical sequences.

We aimed to determine if the syntactic flexibility inherent to the Korean language is mirrored in Korean LMs. Leveraging these unique linguistic characteristics, our objective was to improve the performance of Korean LMs. To this end, we established a methodology for generating Syntactically Incomplete Korean (SIKO) data and subsequently constructed a comprehensive dataset. Further, we evaluated the effectiveness of the SIKO dataset as a resource for fine-tuning LMs, focusing on its practical applicability in enhancing language processing capabilities.

We designed various experiments using the SIKO data. We employed four fundamental tasks used for assessing LM performance: Text Classification (TC), Natural Language Inference (NLI), dialogue topic classification, and dialogue summarization. For the experiments, we used the well-established KLUE benchmark (Park et al., 2021) and the AI-hub data[2] released by national authorities. This compilation includes a wide range of Korean language expressions, from news headlines to conversational texts, thereby expanding our experimental scope to cover the extensive usage of Korean. Additionally, we examined the characteristics and performance across three different Large Language Models (LLMs) with varying sizes and structures, to uncover both commonalities and unique aspects in processing syntactic phenomena.

---

[2]It is a data platform operated by the Korea National Information Society Agency (NIA). It collects and processes artificial intelligence data in six fields and makes it publicly available. `https://www.aihub.or.kr/`

In our study, we observed that expressions featuring syntactic incompleteness are prevalent in real-world Korean language applications. We were able to confirm through inference experiments using `SIKO` data that LLMs are responding to such syntactic incompleteness. Our analysis reveals that the `SIKO` dataset surpasses representative data augmentation methods in efficacy, especially for fine-tuning purposes. Significantly, it demonstrates superior capability in improving the handling of inputs that are syntactically incomplete, a common phenomenon in typical Korean usage.

The contributions of our study are as follows:

- This study is the first to explore syntactically incomplete Korean sentences.
- We constructed the Syntactically Incomplete Korean (`SIKO`) dataset.
- Through downstream testing with `SIKO`, we verified that the Korean LLM exhibits flexibility when dealing with syntactically incomplete inputs.
- Models tuned using `SIKO` demonstrated improved performance in handling incomplete data commonly found in Korean.

## 2 Related works

**Syntactic Information** In neural network-based Natural Language Processing (NLP) research, syntactic information has been a consistently explored topic. Further emphasizing its significance, Vanderwende & Dolan (2005) addressed a considerable number of problems in the PASCAL RTE challenge using solely syntactic information (Dagan et al., 2005). The question of whether these LMs genuinely understand the syntax of human language and align their learning outcomes with human comprehension remains a subject of continued interest (Sinha et al., 2020).

**Word Order** Word order is a prominent element of syntactic information. While research asserting the lack of importance of word order in input data has been presented (Pham et al., 2020), consistent efforts are also being made in studies attempting to enhance performance through training on data with disrupted word order (Sinha et al., 2021). Furthermore, endeavors persist to comprehend how LMs learn human language by utilizing models trained on data with shuffled corpus (Abdou et al., 2022). Korean, while possessing a formal word order, exhibits relative flexibility from it due to the presence of case markers (Lee & Im, 1997). We seek to determine whether the Korean LM also reflects this flexibility.

**Incomplete Sentences for data augmentation** Recent research has demonstrated success in enhancing model robustness and performance by utilizing noisy or incomplete data. Consistency learning, as performed by Zhou et al. (2021) with perturbed data and Gao et al. (2022) with dropout-applied incomplete data, is one approach. Data augmentation methods, such as the repetition method employed by Wu et al. (2022) and the replacement of low-information words proposed by Xie et al. (2020), have also shown promise. General and simple data augmentation methods like Easy data augmentation (EDA) or An easier data augmentation (AEDA) have already become widely used techniques (Wei & Zou, 2019; Karimi et al., 2021). We aimed to augment data by considering the flexibility of word order and case markers as a form of noise.

## 3 Syntactic Flexibility in Korean

We explored the frequency of case marker omissions and word order changes in practical Korean usage. For this investigation, we took 1,000 samples each from two tasks within the KLUE benchmark and conversational data from AI-hub. We assessed the presence of omitted case markers and deviations in word order, and determined whether such phenomena led to ambiguity or challenges in interpreting the meaning of the sentences.

Additionally, we restored omitted case markers and corrected word order in 300 samples for each task. This process allowed us to identify the number of case marker positions,

| Case of Syntactical Completeness | # of samples | KLUE-TC | KLUE-NLI | AI-Hub Dialogue |
|---|---|---|---|---|
| Complete Case Marker | 1,000 | 6.60% | 54.30% | 7.70% |
| Case Marker Omitted & No Translation Issues | 1,000 | 93.40% | 43.70% | 79.30% |
| Case Marker Omitted & Translation Issues Present | 1,000 | 0% | 2.00% | 13.00% |
| Average Number of Case Marker Positions | 300 | 4.23 | 7.15 | 8.86 |
| Case Marker Omission Rate | 300 | 80.10% | 21.57% | 54.25% |
| Canonical Word Order | 1,000 | 88.00% | 95.70% | 65.00% |
| Non-Canonical Word Order & No Translation Issues | 1,000 | 12.00% | 4.30% | 35.00% |
| Non-Canonical Word Order & Translation Issues Present | 1,000 | 0% | 0.00% | 0.00% |
| Word Order Correction Rate (for Non-Canonical Word Order) | 300 | 45.35% | 31.10% | 28.98% |

Table 1: Analyzing Case Marker Omission and Word Order Variations in Korean with KLUE-TC, KLUE-NLI, and AI-hub Dialogue Datasets. **Case Marker Omitted, Non-Canonical Word Order**: Instances with omitted case markers or non-standard word order. **Translation Issues Present**: Ambiguous or uninterpretable sentences. **Number of Case Marker Positions**: The number of sentence case markers after restoration. **Case Marker Omission Rate**: The ratio of omitted to total case markers. **Word Order Correction Rate**: The word-level edit distance ratio of correction effort to total words.

| SIKO case | Description | Example |
|---|---|---|
| original data | Unprocessed data | 나비가 꿀을 마신다. |
| $CM_{del}$ | Remove case markers | 나비__ 꿀__ 마신다. |
| $Shuf_{Sem.Presrv}$ | Rearrange sentence order while maintaining meaning using chatGPT | 꿀을 나비가 마신다. |
| $Shuf_{Sem.Presrv}$&$CM_{del}$ | Remove case markers from $Shuf_{Sem.Presrv}$ data | 꿀__ 나비__ 마신다. |
| $Shuf_{Sem.Non.Presrv}$ | Randomly shuffle the sentence order | 마신다 꿀을 나비가. |
| $Shuf_{Sem.Non.Presrv}$&$CM_{del}$ | Remove case markers from $Shuf_{Sem.Non.Presrv}$ data | 마신다 꿀__ 나비__. |

Table 2: An example of SIKO data from Figure 1. $Shuf_{Sem.Presrv}$ and $Shuf_{Sem.Non.Presrv}$ might yield identical results. Red denotes the subject and its marker, blue the object and its marker, and green the predicate. The __ signifies deleted case markers.

assess omission rates across datasets, and evaluate the scale of word order modifications. The findings are showcased in Table 1. All tasks described in this section were manually performed, and details about the individuals involved are provided in the Experiment Settings Section ( 5.4).

The rate of case marker omission was calculated by comparing the number of omitted case markers to the total number of case markers after their restoration. The word order correction rate was determined by taking the word-level edit distance between the original data and the corrected word order data, then dividing by the total word count in the data.

From this analysis, we ascertained that the omission of case markers and alterations in word order are common in ordinary Korean usage. Specifically, case marker omission occurred in 97.4% of KLUE-TC instances and non-canonical word orders were found in 35% of AI-hub dialogues. The scale of these phenomena is underscored by their extensive occurrence: an average omission rate of 80.1% for case markers in KLUE-TC and an average word order correction rate of 45.35% in KLUE-TC.

Notably, even with such syntactical incompleteness, the sentences largely retained their clarity of meaning. In dialogue data, there were instances where interpretation was challenged due to case marker omissions. However, it's important to note that these were extracted from conversations, potentially lacking some contextual or background information. This investigation reaffirms that while Korean showcases syntactic flexibility, it remains bounded to ensure semantic clarity.

# 4 `SIKO`: Syntactic Incomplete Korean dataset

In Section 3, we demonstrated Korean's extensive syntactic flexibility and its frequent occurrence in actual use. We aimed to determine if Korean LMs reflect this variability. To this end, we generated datasets with deliberate syntactic incompleteness. This approach occasionally led to reductions in syntactic detail and interpretability.

We randomly sampled 22,000 instances from three data types (KLUE-TC, KLUE-NLI, AIhub-dialogue) and developed five variants of `SIKO` data. Of these, 2,000 instances (and their `SIKO` derivatives) served as the test set in Section 6.1, while 20,000 (with their `SIKO` derivatives) were used for fine-tuning in Section 6.2. Additional details on data statistics, prompts, and reviewer information are documented in Appendix B.

### 4.1   Case Markers Deletion ($CM_{del}$)

We removed all case markers from the data to thoroughly eliminate the syntactic information they provide. While partial omission of case markers is common in Korean, complete removal is rare and generally limited to specific contexts, such as news headlines. We crafted this data by analyzing sentence structures with a morphological analyzer and stripping out the components that correspond to case markers.

Morphological analysis is a fundamental task in Korean NLP and is crucial for Korean EDA. The method of removing case markers is as cost-efficient as EDA in terms of data generation. The morphological analyzer we used was developed in-house for the services of the company I am affiliated with. In terms of general performance, it has recorded an F1 score of 98.826 in evaluations conducted on sampled news article sentences.

### 4.2   Semantic Preserving Shuffling ($Shuf_{Sem.Presrv}$)

As mentioned earlier, Korean permits certain variations in word order without altering the meaning. We changed the word order in our dataset while preserving the sentence's meaning. These sentences may not conform to typical Korean patterns. Initial reordering was conducted by ChatGPT, followed by human reviewers ensuring semantic integrity.

### 4.3   Semantic Non-Preserving Shuffling ($Shuf_{Sem.Non.Presrv}$)

We generated data by shuffling word order using Python's Random library, a method distinct from EDA's simple two-word swap, as it involves rearranging words across the entire sentence. This process may diverge significantly from typical Korean structures, potentially compromising syntactic integrity and original meaning.

### 4.4   Mixed Application ($Shuf_{Sem.Presrv}\&CM_{del}$ and $Shuf_{Sem.Non.Presrv}\&CM_{del}$)

We created datasets where we simultaneously applied word order changes and case marker deletion. We employed two distinct methods to change the word order, and then removed case markers from the altered data. This led to the most pronounced reduction in syntactic information throughout our experiments.

## 5   Experiment Setting

### 5.1   Datasets

#### 5.1.1   KLUE Benchmark

The KLUE benchmark, inspired by GLUE (Wang et al., 2018), stands as a premier dataset in Korea. For our study on syntactically incomplete data, we omitted tasks that the answer resides within the input text, such as machine reading comprehension, due to the challenges of uniformly altering word order in both questions and answers. Thus, we selected two tasks from the available eight. The KLUE dataset comprises Training, Validation, and Test segments; our study solely employed the Training subset with accessible labels.

**Text Classification (KLUE-TC)**   The aim of this task is to identify the subject or theme from a given text snippet. The dataset consists of 45,680 news headline titles. It's common for Korean news headlines to minimize the use of case markers. We classified the news

headlines into one of seven categories based on their content: 'politics', 'economy', 'society', 'culture', world', 'IT/science', or 'sports'.

**Natural Language Inference (KLUE-NLI)**  The objective of NLI is to determine the relationship between a premise and a hypothesis. The dataset, formulated in standard literary style, consists of 23,000 entries. Based on a given premise, an NLI model categorizes the hypothesis as either entailment (true), contradiction (false), or neutral (undeterminable).

### 5.1.2 AI-Hub Korean Dialogue Dataset

The AI-Hub dataset, known for its variety in dialogue types such as daily conversations and debates, consists of 279,992 dialogues primarily in the colloquial style typical of messaging platforms. Dialogues usually feature 2 to 5 participants and average 11.27 exchanges per conversation, with each dialogue annotated with a topic and summary.

**Text Classification (Dial-TC)**  This task involves classifying the topic of a given conversation into one of nine categories: 'Personal and Relationships', 'Beauty and Health', 'Commerce (Shopping)', 'Current Affairs and Education', 'Food and Drink', 'Leisure', 'Professions', 'Residential and Living', and 'Events'.

**Summarization (Dial-Sum)**  This task requires generating a summary of a provided conversation, utilizing the same data as the TC task.

### 5.2 Models

**PKO-T5**  The PKO-T5 (Park, 2022), released by PAUST, is a T5 (Raffel et al., 2020) based Korean sequence-to-sequence model using Byte Pair Encoding (BPE) for out-of-vocabulary words. It's trained on sources like Namuewiki, Wikipedia, and Modu Corpus for T5's unsupervised task. PAUST offers three model sizes: small, base, and large. For our experiments, we primarily used the base model and examined the effects of scaling up to the large model.

**Ko-GPT-Trinity 1.2B**  The Ko-GPT-Trinity[3], developed by SK telecom, adapts the GPT-3 architecture (Brown et al., 2020). It was trained on the extensive "ko-DAT" dataset, comprising 35 billion tokens over 72,000 iterations, using a masked autoregressive LM approach with cross-entropy loss for evaluation.We examined its syntactic flexibility in a causal LM framework, where the model's size, like our selection of T5-large, was crucial.

**chatGPT**  ChatGPT (OpenAI, 2022) by OpenAI is a GPT variant tailored for coherent conversational responses. Sourced from a vast internet corpus, it excels in dialogue generation across multiple languages. We employed this model for generating semantically preserved word order alteration data and measuring zero-shot performance on syntactically incomplete inputs. In-Context Learning(ICL) was not used; the tasks were performed purely in zero-shot mode. The prompts and detailed information used are recorded in the Appendix B.3.2.

### 5.3 Implementation Details

We conducted fine-tuning and generation experiments for syntactic flexibility using an NVIDIA A100 40GB GPU. We employed the Adam optimizer with decoupled weight decay ( Kingma & Ba (2014); Loshchilov & Hutter (2017)) and set a learning rate of 5e-5 with a warm-up period spanning 3 epochs. Some of the experiments were conducted in a zero-shot manner, without any tuning or training. Detailed library versions and additional information are available in the Appendix E.1.

---

[3]https://www.sktelecom.com/

| | T5-base | | | | | T5-Large | | | | |
|---|---|---|---|---|---|---|---|---|---|---|
| | KLUE-TC | KLUE-NLI | Dial-TC | Dial-Sum | | KLUE-TC | KLUE-NLI | Dial-TC | Dial-Sum | |
| incompleteness type | Mac-f1 | Acc. | Mac-f1 | ROUGE-1 | ROUGE-2 | Mac-f1 | Acc. | Mac-f1 | ROUGE-1 | ROUGE-2 |
| ordinary Korean | 85.1 | 79.3 | 73.4 | 76.8 | 46.3 | 85.8 | 80.0 | 69.8 | 79.6 | 46.1 |
| $CM_{del}$ | $85.1_{(0.0)}$ | $79.2_{(-0.1)}$ | $73.1_{(-0.3)}$ | $76.5_{(-0.3)}$ | $44.2_{(-2.1)}$ | 85.7 | $79.6_{(-0.4)}$ | $69.5_{(-0.3)}$ | $78.5_{(-1.1)}$ | $44.2_{(-1.9)}$ |
| $Shuf_{Sem.Presrv}$ | $84.0_{(-1.1)}$ | $75.5_{(-3.8)}$ | $72.8_{(-0.6)}$ | $79.1_{(2.3)}$ | $45.9_{(-0.4)}$ | $84.9_{(-0.8)}$ | $75.8_{(-4.2)}$ | $70.2_{(0.4)}$ | $80.0_{(0.4)}$ | $46.0_{(-0.1)}$ |
| $Shuf_{Sem.Presrv}$ & $CM_{del}$ | $83.9_{(-1.2)}$ | $74.2_{(-5.1)}$ | $72.2_{(-1.2)}$ | $78.5_{(1.7)}$ | $45.7_{(-0.6)}$ | $85.3_{(-0.5)}$ | $75.0_{(-5.0)}$ | $70.5_{(0.7)}$ | $79.6_{(0.0)}$ | $43.6_{(-2.4)}$ |
| $Shuf_{Sem.Non.Presrv}$ | $84.0_{(-1.1)}$ | $71.0_{(-8.3)}$ | $73.3_{(-0.1)}$ | $79.1_{(2.3)}$ | $43.7_{(-2.6)}$ | $84.6_{(-1.2)}$ | $73.5_{(-6.6)}$ | $70.6_{(0.8)}$ | $77.9_{(-1.7)}$ | $44.1_{(-2.0)}$ |
| $Shuf_{Sem.Non.Presrv}$ & $CM_{del}$ | $84.2_{(-0.9)}$ | $69.7_{(-9.6)}$ | $72.7_{(-0.7)}$ | $79.0_{(2.2)}$ | $43.4_{(-2.9)}$ | $84.4_{(-1.4)}$ | $72.1_{(-7.9)}$ | $70.3_{(0.5)}$ | $79.4_{(-0.2)}$ | $42.6_{(-3.5)}$ |

| | KO-GPT | | | | | chatGPT | | | | |
|---|---|---|---|---|---|---|---|---|---|---|
| | KLUE-TC | KLUE-NLI | Dial-TC | Dial-Sum | | KLUE-TC | KLUE-NLI | Dial-TC | Dial-Sum | |
| incompleteness type | Mac-f1 | Acc. | Mac-f1 | ROUGE-1 | ROUGE-2 | Mac-f1 | Acc. | Mac-f1 | ROUGE-1 | ROUGE-2 |
| ordinary Korean | 82.8 | 72.4 | 67.3 | 72.0 | 41.7 | 80.9 | 57.4 | 39.1 | 26.8 | 6.5 |
| $CM_{del}$ | $82.0_{(-0.7)}$ | $70.2_{(-2.1)}$ | $66.5_{(-0.8)}$ | $71.0_{(-1)}$ | $39.9_{(-1.7)}$ | $80.5_{(-0.4)}$ | $57.0_{(-0.4)}$ | $40.5_{(1.4)}$ | $28.3_{(1.5)}$ | $6.6_{(0.1)}$ |
| $Shuf_{Sem.Presrv}$ | $82.5_{(-0.2)}$ | $67.6_{(-4.8)}$ | $68.2_{(0.9)}$ | $72.3_{(0.3)}$ | $41.6_{(-0.1)}$ | $79.7_{(-1.2)}$ | $52.5_{(-4.9)}$ | $39.7_{(0.6)}$ | $27.1_{(0.3)}$ | $6.0_{(-0.5)}$ |
| $Shuf_{Sem.Presrv}$ & $CM_{del}$ | $82.6_{(-0.2)}$ | $65.3_{(-7.1)}$ | $68.3_{(1.0)}$ | $72.0_{(0.0)}$ | $39.5_{(-2.2)}$ | $80.0_{(-0.9)}$ | $51.3_{(-6.1)}$ | $38.4_{(-0.7)}$ | $25.4_{(-1.4)}$ | $6.2_{(-0.3)}$ |
| $Shuf_{Sem.Non.Presrv}$ | $82.2_{(-0.6)}$ | $54.1_{(-18.3)}$ | $68.6_{(1.3)}$ | $70.4_{(-1.5)}$ | $39.9_{(-1.8)}$ | $78.7_{(-2.2)}$ | $38.9_{(-18.5)}$ | $38.8_{(-0.3)}$ | $24.8_{(-2)}$ | $4.9_{(-1.6)}$ |
| $Shuf_{Sem.Non.Presrv}$ & $CM_{del}$ | $81.6_{(-1.2)}$ | $53.7_{(-18.7)}$ | $68.0_{(0.7)}$ | $71.8_{(-0.2)}$ | $38.5_{(-3.1)}$ | $78.9_{(-2)}$ | $42.0_{(-15.4)}$ | $39.7_{(0.6)}$ | $26.1_{(-0.7)}$ | $5.6_{(-0.9)}$ |

Table 3: When SIKO data is input into a model pre-trained or fine-tuned on standard Korean data, we observe the processing outcomes. We report scores for SIKO test cases and their variation from standard Korean cases as score$_{diff}$. The performance for TC was measured using Macro-F1, for NLI using Accuracy, and for summarization using ROUGE-1 and ROUGE-2.

## 5.4 Human-Constructed Data

In this study, multiple human-centric tasks such as evaluation, restoration, and correction were conducted. In Section 3, they identified the presence of omitted case markers or altered word order in general Korean usage and performed the task of restoring omitted case markers or word order to calculate their scale. In Section 4.2, they reviewed and corrected the meaning-preserved word order change data generated by ChatGPT to ensure that the meaning was indeed preserved. Two people directly handled these tasks. Subsequent to their efforts, another individual, termed the reviewer, carried out checks and verifications. All participants brought to the table a minimum of 18 months' experience in creating and evaluating data for NLP. More detailed information on the task guidelines, training methods, and other aspects provided to these individuals is documented in the Appendix B.3.1.

# 6 Experiments

## 6.1 Experiment 1: The Influence of Incomplete Syntax in Inference

We explored how LMs process inputs missing case markers or with non-standard word orders, selecting 2,000 samples from four tasks' training data as our test set. Using the SIKO approach from Section 4, we created five types of syntactically incomplete test sets. The remaining ordinary data was divided in a 9:1 ratio for training and validation during fine-tuning. For TC and NLI tasks with class labels, our sampling preserved original class distributions. We evaluated the inference performance of a LM fine-tuned on typical data with syntactically incomplete test set. We also attempted zero-shot inference with chatGPT to note processing differences. (Unfortunately, T5 and KO-GPT were unable to correctly analyze the task under zero-shot conditions. Therefore, we compared the inference performance using fine-tuned models.)

### 6.1.1 Result

Table 3 presents our experimental results. Notably, case marker deletion ($CM_{del}$) barely affected task performance. However, tasks reacted differently to word order changes, with the NLI task being particularly sensitive. Except for NLI, alterations in word order, whether semantically preserved ($Shuf_{Sem.Presrv}$) or not ($Shuf_{Sem.Non.Presrv}$), showed minimal performance influence. This indicates that grammatical variations minimally affect understanding in contexts like news headlines or informal speech, or that LLMs can effectively interpret their meaning.

| Task | Score | Aug. rate | BASELINE | | | | | SIKO | | | | |
|---|---|---|---|---|---|---|---|---|---|---|---|---|
| | | | non-Aug. | duplication | repetition | AEDA | EDA | $CM_{del}$ | $Shuf_{Sem.Presrv}$ | $Shuf_{Sem.Presrv}$ & $CM_{del}$ | $Shuf_{Sem.Non.Presrv}$ | $Shuf_{Sem.Non.Presrv}$ & $CM_{del}$ |
| KLUE-TC | macro F1 | 0.1 | 81.6 | $82.0_{(1.6)}$ | $81.1_{(0.8)}$ | $80.6_{(2.3)}$ | $83.3_{(0.4)}$ | $83.7_{(0.7)}$ | $\mathbf{84.4}_{(0.5)}$ | $83.6_{(0.7)}$ | $83.1_{(1.0)}$ | $83.3_{(1.5)}$ |
| | | 0.3 | | $82.2_{(0.2)}$ | $83.0_{(3.8)}$ | $82.9_{(0.9)}$ | $83.2_{(0.5)}$ | $83.6_{(0.4)}$ | $\mathbf{84.0}_{(0.8)}$ | $83.4_{(1.4)}$ | $83.5_{(0.8)}$ | $81.8_{(0.7)}$ |
| | | 0.5 | | $82.7_{(0.6)}$ | $83.9_{(0.7)}$ | $80.9_{(3)}$ | $82.2_{(3.5)}$ | $\mathbf{84.5}_{(0.3)}$ | $84.3_{(0.6)}$ | $83.4_{(0.6)}$ | $82.5_{(0.7)}$ | $82.9_{(0.6)}$ |
| | | 1 | | $83.1_{(0.2)}$ | $83.0_{(0.6)}$ | $80.1_{(2.0)}$ | $82.4_{(0.7)}$ | $\mathbf{83.5}_{(0.6)}$ | $83.0_{(0.9)}$ | $83.5_{(0.2)}$ | $80.1_{(3.8)}$ | $82.2_{(1.1)}$ |
| KLUE-NLI | Acc. | 0.1 | 79.7 | $80.5_{(1.2)}$ | $82.9_{(1.6)}$ | $81.6_{(2.2)}$ | $82.1_{(1.6)}$ | $\mathbf{83.5}_{(0.4)}$ | $83.1_{(1.7)}$ | $81.7_{(1.8)}$ | $81.9_{(1.6)}$ | $82.9_{(1.2)}$ |
| | | 0.3 | | $81.2_{(1.2)}$ | $83.2_{(0.4)}$ | $82.6_{(1.0)}$ | $82.3_{(1.3)}$ | $\mathbf{83.8}_{(0.3)}$ | $83.8_{(0.5)}$ | $83.3_{(0.6)}$ | $82.8_{(1.1)}$ | $83.3_{(1.4)}$ |
| | | 0.5 | | $82.2_{(0.4)}$ | $82.3_{(0.6)}$ | $82.2_{(0.7)}$ | $81.6_{(0.5)}$ | $82.5_{(0.9)}$ | $82.6_{(0.7)}$ | $82.3_{(0.7)}$ | $\mathbf{82.6}_{(0.2)}$ | $82.0_{(0.0)}$ |
| | | 1 | | $82.0_{(0.1)}$ | $83.6_{(0.6)}$ | $84.3_{(0.1)}$ | $83.3_{(1.1)}$ | $\mathbf{84.5}_{(0.4)}$ | $83.3_{(1.3)}$ | $83.5_{(0.3)}$ | $83.3_{(0.0)}$ | $84.1_{(0.4)}$ |
| Dial. TC | macro F1 | 0.1 | 64.6 | $67.3_{(0.9)}$ | $69.3_{(2.5)}$ | $69.9_{(1.9)}$ | $69.4_{(1.7)}$ | $69.0_{(0.6)}$ | $70.2_{(2.4)}$ | $69.6_{(2.4)}$ | $69.1_{(1.9)}$ | $\mathbf{70.7}_{(1.9)}$ |
| | | 0.3 | | $67.8_{(1.4)}$ | $68.1_{(2.3)}$ | $69.3_{(2.9)}$ | $69.5_{(1.4)}$ | $68.0_{(1.3)}$ | $\mathbf{72.0}_{(1.3)}$ | $68.9_{(1.5)}$ | $67.7_{(1.2)}$ | $70.8_{(1.2)}$ |
| | | 0.5 | | $68.7_{(1.3)}$ | $68.6_{(1.9)}$ | $69.6_{(2.8)}$ | $69_{(2.3)}$ | $69.2_{(1.4)}$ | $69.1_{(1.4)}$ | $\mathbf{70.6}_{(1.3)}$ | $70.2_{(2.4)}$ | $70.2_{(2.4)}$ |
| | | 1 | | $68.4_{(2.8)}$ | $69.4_{(1.7)}$ | $65.2_{(7.6)}$ | $69.1_{(1.6)}$ | $70.0_{(1.9)}$ | $\mathbf{71.2}_{(0.5)}$ | $69.0_{(2.7)}$ | $71.1_{(2.2)}$ | $69.2_{(2.2)}$ |
| Dial. Sum. | ROUGE-1 | 0.1 | 76.3 | $77.0_{(0.4)}$ | $77.1_{(0.8)}$ | $77.9_{(0.5)}$ | $77.9_{(0.4)}$ | $\mathbf{78.7}_{(0.9)}$ | $78.2_{(0.5)}$ | $77.9_{(0.5)}$ | $77.8_{(0.5)}$ | $78.1_{(0.4)}$ |
| | | 0.3 | | $77.0_{(0.4)}$ | $77.8_{(1.0)}$ | $77.3_{(0.7)}$ | $77.1_{(0.7)}$ | $\mathbf{78.7}_{(0.7)}$ | $77.3_{(1.1)}$ | $77.0_{(0.7)}$ | $78.1_{(0.9)}$ | $78.0_{(0.7)}$ |
| | | 0.5 | | $77.2_{(0.8)}$ | $77.7_{(0.2)}$ | $77.1_{(1.4)}$ | $\mathbf{78.4}_{(0.4)}$ | $78.3_{(0.6)}$ | $78.3_{(0.6)}$ | $78.0_{(0.5)}$ | $77.9_{(0.6)}$ | $77.8_{(0.8)}$ |
| | | 1 | | $77.5_{(0.5)}$ | $77.7_{(0.4)}$ | $78.5_{(0.5)}$ | $78.2_{(0.7)}$ | $78.2_{(0.3)}$ | $77.5_{(0.2)}$ | $\mathbf{78.9}_{(2.0)}$ | $77.9_{(0.1)}$ | $78.2_{(0.0)}$ |
| | ROUGE-2 | 0.1 | 44.9 | $43.9_{(1.4)}$ | $44.1_{(0.8)}$ | $45.0_{(0.0)}$ | $44.5_{(0.1)}$ | $45.3_{(0.6)}$ | $44.9_{(0.7)}$ | $\mathbf{45.4}_{(1.0)}$ | $44.4_{(0.2)}$ | $44.7_{(0.8)}$ |
| | | 0.3 | | $44.2_{(0.2)}$ | $\mathbf{45.4}_{(0.2)}$ | $44.4_{(0.6)}$ | $44.7_{(0.3)}$ | $44.6_{(0.3)}$ | $44.3_{(0.1)}$ | $43.2_{(0.3)}$ | $45.3_{(0.5)}$ | $43.7_{(0.5)}$ |
| | | 0.5 | | $44.5_{(1.2)}$ | $44.2_{(0.4)}$ | $43.5_{(1.1)}$ | $44.1_{(0.5)}$ | $45.2_{(0.4)}$ | $45.2_{(0.9)}$ | $\mathbf{45.5}_{(1.2)}$ | $44.3_{(0.6)}$ | $43.9_{(0.3)}$ |
| | | 1 | | $44.7_{(0.5)}$ | $44.0_{(1.0)}$ | $44.6_{(1.6)}$ | $44.0_{(0.2)}$ | $44.7_{(0.6)}$ | $44.3_{(1.7)}$ | $43.1_{(1.7)}$ | $43.8_{(0.9)}$ | $\mathbf{45.4}_{(1.8)}$ |

Table 4: Data tuning results using SIKO data were compared to representative augmentation strategies across transformer architectures for downstream performance. Tests varied augmentation ratios and were repeated five times, with results in $mean_{std}$ format.

This pattern persisted with larger models (e.g., t5-large) and different architectures (e.g., KO-GPT). In a zero-shot context, chatGPT showed similar performance trends despite lacking prior information. Without fine-tuning, chatGPT's performance lagged behind other models, struggling especially in summarization tasks by generating verbose sentences, leading to low Rouge scores.

## 6.2 Experiment 2: Training with SIKO

To assess the SIKO impact on training, we initially restricted our fine-tuning data to 20,000 instances per task, due to data construction and validation capabilities. As described in Section 4, we generated five SIKO data variants by incorporating 10%, 30%, 50%, and 100% of ordinary instances. We fine-tuned models with a blend of 20,000 ordinary instances and SIKO data, comparing performance to other augmentation methods to establish its training efficacy. Performance was evaluated on a ordinary test set (excluding SIKO modifications) to validate SIKO as an augmentation technique. Following Yoo et al. (2021), this procedure was repeated five times to analyze average performance and variability. This verification method was employed to avoid sampling bias.

### 6.2.1 Baselines: Augmentation

We compared its performance using the following four representative data augmentation methods, with detailed descriptions of these methods documented in the appendix D.

**Duplicate** - This method simply adds the sampled data to the tuning data without any processing. It serves as a baseline, highlighting performance improvements attributable solely to increased data volume.

**Repetition** - Introduced by Wu et al. (2022), this approach duplicates a randomly selected word within the data.

**EDA** - Suggested by Wei & Zou (2019), this technique involves four probabilistic actions applied to the data: synonym insertion, synonym substitution, random deletion, and position swapping.

**AEDA** - As presented by Karimi et al. (2021), this method involves the probabilistic insertion of special characters into the data.

| completeness cases | KLUE-TC | | | KLUE-NLI | | | Dialogue TC | | |
|---|---|---|---|---|---|---|---|---|---|
| | Macro-F1 | | | Acc. | | | Macro-F1 | | |
| | whole | Complete case marker | **Incomplete case marker** | whole | Complete case marker | **Incomplete case marker** | whole | Complete case marker | **Incomplete case marker** |
| rate | 100.0 | 6.6 | 93.4 | 100.0 | 54.3 | 45.7 | 100.0 | 7.7 | 92.3 |
| baseline | | | | | | | | | |
| non-augmented | 81.6 | 94.0 | 80.7 | 79.7 | 79.7 | 79.6 | 64.6 | 68.4 | 64.3 |
| duplication | 82.3 | 94.0 | 81.5 | 82.4 | 84.8 | 79.6 | 68.4 | 71.5 | 68.2 |
| repetition | 83.5 | 96.0 | 82.6 | 81.4 | 83.0 | 79.6 | 69.2 | 71.7 | 69.0 |
| EDA | 82.0 | 92.2 | 81.3 | 81.4 | 81.7 | 81.1 | 68.9 | 71.5 | 68.7 |
| SIKO | | | | | | | | | |
| $CM_{del}$ | 84.4 | 98.0 | **83.4** | 83.0 | 83.4 | **82.6** | 69.7 | 70.7 | 69.7 |
| $Shuf_{Sem.Presrv}$ | 84.2 | 98.0 | 83.2 | 81.8 | 82.1 | 81.3 | 70.2 | 72.3 | **70.0** |
| $Shuf_{Sem.Non.Presrv}$ | 84.0 | 100.0 | 82.9 | 82.0 | 83.9 | 79.8 | 69.4 | 71.9 | 69.2 |

Table 5: 50% augmentation test results with T5-base by case marker completeness. $CM_{del}$ method enhances performance on incomplete case marker data.

| completeness cases | KLUE-TC | | | KLUE-NLI | | | Dialogue TC | | |
|---|---|---|---|---|---|---|---|---|---|
| | Macro-F1 | | | Acc. | | | Macro-F1 | | |
| | whole | Complete word order | **Incomplete word order** | whole | Complete word order | **Incomplete word order** | whole | Complete word order | **Incomplete word order** |
| rate | 100.0 | 88.0 | 12.0 | 100.0 | 95.7 | 4.3 | 100.0 | 65.0 | 35.0 |
| baseline | | | | | | | | | |
| non-augmented | 81.6 | 81.3 | 83.5 | 79.7 | 80.7 | 56.9 | 64.6 | 64.9 | 64.4 |
| duplication | 82.3 | 82.9 | 78.0 | 82.4 | 83.6 | 56.5 | 68.4 | 70.2 | 67.5 |
| repetition | 83.5 | 84.7 | 74.6 | 81.4 | 82.3 | 63.3 | 69.2 | 70.3 | 68.6 |
| EDA | 82.0 | 82.4 | 78.9 | 81.4 | 82.2 | 65.5 | 68.9 | 70.4 | 68.2 |
| SIKO | | | | | | | | | |
| $CM_{del}$ | 84.4 | 84.4 | 84.5 | 83.0 | 84.0 | 60.7 | 69.7 | 70.7 | 69.2 |
| $Shuf_{Sem.Presrv}$ | 84.2 | 83.9 | **86.0** | 81.8 | 82.4 | **68.1** | 70.2 | 71.1 | **69.7** |
| $Shuf_{Sem.Non.Presrv}$ | 84.0 | 84.1 | 83.8 | 82.0 | 82.7 | 66.9 | 69.4 | 69.6 | 69.3 |

Table 6: 50% Augmentation test results with T5-base by word order completeness. **Shuf** based methods boost performance on data with altered Word Order.

### 6.2.2 Result

Table 4 presents our fine-tuning experiment results, highlighting consistent performance improvements with SIKO data in most scenarios. Notably, deleting case markers and altering word order ($Shuf_{Sem.*}$ & $CM_{del}$) resulted in inconsistent performance changes, from minor to major improvements, likely from losing syntactic information. Due to the use of syntactically noisy data, overly high SIKO data proportions (e.g., 100% augmentation ratio) led to reduced gains in performance.

Integrating insights from Section 3, we evaluated SIKO's impact on performance. Table 5 details results from a 50% augmentation experiment by case marker usage completeness, while Table 6 focuses on sentence order completeness. Models fine-tuned on $CM_{del}$ data exhibited high performance in handling incomplete case markers. Similarly, models adjusted with $Shuf_{Sem.}$ data demonstrated superior performance in scenarios with altered sentence order. Interestingly, while KLUE-TC showed large performance improvements with perfect case marker usage, this appears to be due to the rare occurrence of such usage (6.6%), suggesting that the performance changes were significantly influenced by a few test cases.

Section 3 highlighted the prevalence of syntactically incomplete sentences in Korean, with SIKO effectively enhancing model performance on such sentences. As Section 4 elaborates, strategies like $CM_{del}$ and $Shuf_{Sem.Non.Presrv}$ provide cost-efficient data augmentation alternatives to EDA, demonstrating consistent performance improvements and establishing their reliability. To achieve a precise analysis, we focused our attention on test sets with unambiguously defined correct answers, as seen in Tables 5 and 6. We excluded the summary test set from this comparison because Rouge score could be influenced by the length or expressiveness of the generated sentence. Furthermore, for a streamlined comparison, we omitted mixed-generation methods and AEDA due to its conceptual overlap with EDA. A comprehensive table that includes these excluded components can be found in the Appendix C.2.

Furthermore, to evaluate the effects on models beyond the T5-base, we ran experiments with 50% augmented data on both the GPT-3 based SKT-Kogpt-trinity model and the T5-large model. While the boost in performance from augmentation was subtle, a consistent pattern emerged. More detailed results and analysis are recorded in the appendix C.1.

## 7  Discussion

In this study, we established that Korean's syntactic flexibility are indeed mirrored in LMs. We also demonstrated that data generated via the SIKO method enhances comprehension of ordinary Korean. The scope of our research was intentionally limited to Korean, constrained by our linguistic expertise and the experimental scope. Yet, it's noteworthy that numerous languages, including Japanese, certain Turkic languages, Mongolian, and Hungarian, possess case markers or exhibit flexibility in sentence structure, like Japanese, Russian, and Latin. We anticipate that our findings will significantly benefit LMs developed for these languages.

Our methodology leveraged training data infused with syntactic noise to boost model performance. By experimenting with various rates of data augmentation, we ascertained the necessity for moderation in augmentation to avoid detrimental effects. Future studies will aim to pinpoint the optimal augmentation level, evaluate potential drawbacks, and assess the method's utility across different tasks.

## 8  Conclusion

We sought to determine whether Korean LMs reflect the inherent syntactic flexibility of the Korean language, especially regarding case marker usage and word order. Our initial findings confirmed the frequent use of incomplete case markers and word orders in Korean through publicly available benchmark datasets. To delve deeper, we introduced the Syntactically Incomplete Korean (SIKO) dataset. Through SIKO, it became evident that Korean LMs can flexibly handle syntactically incomplete inputs. Fine-tuning with SIKO enhances this capability to process common incomplete syntactic forms in Korean. Moreover, the dataset's low construction cost, coupled with significant performance enhancements, solidifies its standing as an effective data augmentation technique.

## Acknowledgement

This work was supported by Institute of Information & communications Technology Planning & Evaluation (IITP) grant funded by the Korea government(MSIT) [No.2022-0-00641, XVoice: Multi-Modal Voice Meta Learning]

This work has supported by the National Research Foundation of Korea(NRF) grant funded by the Korea government(MSIT)(No. 2022R1F1A1071047).

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

# A   Licenses

We list the licenses of each source dataset that we utilized in the creation of SIKO.

- KLUE: Creative Commons Attribution-ShareAlike 4.0 International License.
- AI-hub dialogue: This data is modifiable, but redistribution is not allowed.

# B   SIKO and Resources for Construction

|  | KLUE-TC | KLUE-NLI | DIALOGUE |
|---|---|---|---|
| # of total data | 45,680 | 23,000 | 279,992 |
| # of training data for 4.3.1 | 43,680 | 20,998 | 277,992 |
| # of test data for 4.3.1 | 2,000 | 2,000 | 2,000 |
| # of training data for 4.3.2 | 20,000 | 20,000 | 20,000 |
| # of test data for 4.3.2 | 2,000 | 2,000 | 2,000 |
| average of length | 27.37 | 45.41/24.92 | 120.62 |
| average of words | 6.61 | 10.74/5.84 | 27.87 |
| # of category | 7 | - | 9 |
| average of speaker number | - | - | 2.07 |
| average of turns | - | - | 11.27 |
| average of length in each turn | - | - | 10.7 |
| average of words in each turn | - | - | 2.47 |

Table 7: Benchmark data information

|  | Training Data | Validation Data | Test Data |
|---|---|---|---|
| $CM_{del}$ | 18000 | 2000 | 2000 |
| $Shuf_{Sem.Presrv}$ | 18000 | 2000 | 2000 |
| $Shuf_{Sem.Presrv}$ & $CM_{del}$ | 18000 | 2000 | 2000 |
| $Shuf_{Sem.Non.Presrv}$ | 18000 | 2000 | 2000 |
| $Shuf_{Sem.Non.Presrv}$ & $CM_{del}$ | 18000 | 2000 | 2000 |

Table 8: Generated SIKO Data Statistics

## B.1   SIKO statistics

Tables 7 and 8 record the statistics of each benchmark dataset used for this experiment and the statistics of the generated SIKO data, respectively.

## B.2   Evaluation Task Information

### B.2.1   KLUE Benchmark

The KLUE (Korean Language Understanding Evaluation) benchmark is a comprehensive suite of tasks designed to evaluate and advance the capabilities of language models specifically in understanding the Korean language. It includes a variety of tasks such as text classification, named entity recognition, semantic textual similarity, natural language inference, and question answering. These tasks cover a wide range of linguistic phenomena and challenges that are representative of real-world language use in the Korean context. The creation of KLUE aims to foster the development of AI models that better understand and process the Korean language, contributing to the global progress in natural language processing technologies.

### B.2.2 AI-hub Data

AI-hub is a comprehensive data platform in South Korea, operated under the auspices of the National Information Society Agency (NIA). Its primary mission is to facilitate advancements in artificial intelligence by offering an extensive repository of high-quality datasets. These datasets span a wide range of fields, including language, voice, image, and more, catering to diverse AI research and development needs. The platform serves as a critical resource for developers, researchers, and businesses engaged in AI, providing them with the necessary data to train, test, and refine their models and applications. By fostering innovation and supporting the AI ecosystem in Korea, AI-hub plays a pivotal role in driving the country's technological advancement and competitiveness in the global AI landscape.

### B.3 Data Generation and Inspection

#### B.3.1 Human Inspection

This experiment involved a total of three human inspectors. They are individuals contracted on an annual basis by our company to build NLP data, generate natural language data, create tagged corpora, or evaluate model performance. These professionals have over 18 months of experience and hold at least a bachelor's degree. The team consisted of two workers and one reviewer, where the workers performed the tasks below, and the reviewer checked and corrected their outputs.

We requested them to undertake the following two tasks:

**Review of word order alteration data** : We constructed draft data for semantically preserved word order alterations using chatGPT. The workers reviewed the results to ensure that 1) the meaning was preserved, and 2) all words from the original sentence were retained. If either condition was not met, the workers would reconstruct it themselves.

**Statistical survey on the use of syntactically incomplete Korean** : To compile the statistics in Table 1, workers checked 1) the case of omitted case markers, 2) the restoration of omitted case markers, 3) the case markers in sentences with incorrect word order, 4) the generation of sentences with correct word order, and 5) the determinability of interpretation.

#### B.3.2 chatGPT

We utilized chatGPT in constructing semantically preserved word order alteration data. Table 9 translates the prompt we used into English. Figure 2 documents the actual code and version information used. Figure 3 displays the result when the prompt is entered into the Web UI, featuring the same example sentence as in Figure 1.

| version | 'deploy-ranker-gpt35turbo engine' (May 2023 version). |
|---|---|
| prompt | Please change the word order of the following Input sentence without altering its meaning. Each word in the sentence be maintained as is.

Provide a short answer.

Input sentence: {DATA}

What is the altered sentence? |
| Korean prompt | 다음 문장의 의미가 변하지 않게 어순을 변경해줘
문장의 각 단어는 그대로 유지되어야 해

입력: {DATA}

변경된 문장은? |

Table 9: ChatGPT information. Version & Translated Prompt

## C  Additional Experiments

### C.1  Augmentation Result using Large Model

Furthermore, to evaluate the effects on models beyond the T5-base, we ran experiments with 50% augmented data on both the GPT-3 based SKT-Kogpt-trinity model and the T5-large model. While the boost in performance from augmentation was subtle, a consistent pattern emerged. When contrasting the outputs of models that were fine-tuned without any data augmentation, models with more parameters consistently surpassed the T5-base. This observation suggests that larger models inherently possess greater syntactic flexibility than their smaller counterparts. The fact that both large and small models exhibit nearly identical performance ceilings when fine-tuned with augmented data supports this interpretation.

### C.2  Omitted Experimental Results

Table 12 and table 11 are the full versions of Table 5 and table 6.

## D  Baseline: Augmentation Methods

**Duplicate**    - This method simply adds the sampled data to the tuning data without any processing. It serves as a baseline, highlighting performance improvements attributable solely to increased data volume.

**Repetition**    - Introduced by Wu et al. (2022), this approach duplicates a randomly selected word within the data.

**EDA**    - Easy Data Augmentation (EDA) is a set of straightforward techniques aimed at enhancing natural language processing (NLP) models by augmenting textual data. Introduced by Wei & Zou (2019), EDA improves model performance through four operations: synonym replacement, random insertion of synonyms, random swapping of words, and random deletion of words in sentences. These methods increase the diversity of training data by slightly altering sentences, thereby helping models become more robust and generalize better. EDA is especially valuable for projects with limited labeled data, offering an effective way to expand datasets without significantly altering their original meaning. We used the 'koeda[4]' library to generate EDA and AEDA data, and this library references the Korean WordNet.

**AEDA**    - As presented by Karimi et al. (2021), A lighter Easier Data Augmentation, is a data augmentation technique for NLP that focuses on simplifying and streamlining the augmentation process. Contrary to its predecessor, EDA, which involves synonym replacement, random insertion, swapping, and deletion, AEDA simplifies the augmentation by predominantly using random insertion of punctuation marks into text data. This method aims to enhance model performance by diversifying the training data with minimal changes, thereby maintaining the semantic integrity of the original text while introducing syntactic variations. AEDA's approach is beneficial for improving the robustness of NLP models, particularly in scenarios with limited training data, by offering a straightforward and effective means to augment text data.

## E  Implementation Details

### E.1  Development Environment

Table 13 shows our development environments. We conducted fine-tuning and generation experiments for syntactic flexibility using an NVIDIA A100 40GB GPU 8core. We employed

---

[4]https://github.com/toriving/KoEDA

the Adam optimizer with decoupled weight decay ( Kingma & Ba (2014); Loshchilov & Hutter (2017)) and set a learning rate of 5e-5 with a warm-up period spanning 3 epochs.

| | | | BASELINE | | | | SIKO | | | | |
|---|---|---|---|---|---|---|---|---|---|---|---|
| Task | Score | aug rate | duplication | repetition | AEDA | EDA | $CM_{del}$ | $Shuf_{Sem.Presrv}$ | $Shuf_{Sem.Presrv}$ & $CM_{del}$ | $Shuf_{Sem.Non.Presrv}$ | $Shuf_{Sem.Non.Presrv}$ & $CM_{del}$ |
| KLUE-TC | macro-F1 | 0.5 | $82.9_{(1.1)}$ | $83.0_{(1.0)}$ | $84.4_{(0.7)}$ | $83.1_{(2.5)}$ | $84.4_{(1.0)}$ | $84.9_{(0.7)}$ | $83.7_{(1.0)}$ | $80.5_{(3.6)}$ | $84.0_{(0.5)}$ |
| KLUE-NLI | Acc. | 0.5 | $81.1_{(2.6)}$ | $82.5_{(1.0)}$ | $83.1_{(1.8)}$ | $82.5_{(3.9)}$ | $83.8_{(2.6)}$ | $84.7_{(2.2)}$ | $84.7_{(0.8)}$ | $80.1_{(6.9)}$ | $83.5_{(1.0)}$ |
| Dial. TC | macro-F1 | 0.5 | $67.9_{(0.6)}$ | $68.8_{(1.0)}$ | $68.6_{(1.6)}$ | $69.0_{(0.7)}$ | $69.0_{(1.2)}$ | $70.3_{(1.7)}$ | $70.1_{(1.3)}$ | $69.1_{(1.9)}$ | $68.1_{(1.8)}$ |
| Dial. Sum. | ROUGE-1 | 0.5 | $77.8_{(0.7)}$ | $78.7_{(0.7)}$ | $78.9_{(0.3)}$ | $79.8_{(0.6)}$ | $79.6_{(0.5)}$ | $79.7_{(0.2)}$ | $79.7_{(0.3)}$ | $78.8_{(0.9)}$ | $78.6_{(0.8)}$ |
| | ROUGE-2 | 0.5 | $42.9_{(0.5)}$ | $44.8_{(1.5)}$ | $45.5_{(1.1)}$ | $44.7_{(0.7)}$ | $44.6_{(1.2)}$ | $45.1_{(1.7)}$ | $45.0_{(1.0)}$ | $44.0_{(1.4)}$ | $43.0_{(0.7)}$ |

Results of the 50% expansion test using the T5-large model

| | | | BASELINE | | | | SIKO | | | | |
|---|---|---|---|---|---|---|---|---|---|---|---|
| Task | Score | aug rate | duplication | repetition | AEDA | EDA | $CM_{del}$ | $Shuf_{Sem.Presrv}$ | $Shuf_{Sem.Presrv}$ & $CM_{del}$ | $Shuf_{Sem.Non.Presrv}$ | $Shuf_{Sem.Non.Presrv}$ & $CM_{del}$ |
| KLUE-TC | macro-F1 | 0.5 | $83.8_{(0.3)}$ | $83.8_{(0.4)}$ | $83.8_{(1.0)}$ | $83.9_{(0.9)}$ | $83.7_{(0.2)}$ | $84.2_{(0.4)}$ | $83.9_{(0.8)}$ | $84.0_{(0.5)}$ | $83.3_{(0.2)}$ |
| KLUE-NLI | Acc. | 0.5 | $82.2_{(0.4)}$ | $82.8_{(1.1)}$ | $82.6_{(0.9)}$ | $80.8_{(1.2)}$ | $83.8_{(0.5)}$ | $83.2_{(1.0)}$ | $82.9_{(0.6)}$ | $82.4_{(1.8)}$ | $82.9_{(0.4)}$ |
| Dial. TC | macro-F1 | 0.5 | $67.1_{(1.8)}$ | $66.2_{(0.7)}$ | $69.7_{(2.2)}$ | $70.9_{(0.7)}$ | $70.0_{(4.4)}$ | $70.3_{(2)}$ | $69.8_{(4)}$ | $67.9_{(3.8)}$ | $68.1_{(2)}$ |
| Dial. Sum. | ROUGE-1 | 0.5 | $78.0_{(0.5)}$ | $78.7_{(1.0)}$ | $77.8_{(0.6)}$ | $78.2_{(0.4)}$ | $78.9_{(0.6)}$ | $77.1_{(1.0)}$ | $78.8_{(0.5)}$ | $78.0_{(2.7)}$ | $78.1_{(0.8)}$ |
| | ROUGE-2 | 0.5 | $44.9_{(0.8)}$ | $41.9_{(1.6)}$ | $44.2_{(1.6)}$ | $45.2_{(0.9)}$ | $42.3_{(1.5)}$ | $43.9_{(0.4)}$ | $43.5_{(2.4)}$ | $40.7_{(1.8)}$ | $44.2_{(0.3)}$ |

Results of the 50% expansion test using the KO-GPT

Table 10: Results from the 50% augmentation test with T5-large and GPT3. Despite size and structure differences, methods using SIKO data consistently improved performance, echoing the T5-base results.

| | KLUE-TC | | | KLUE-NLI | | | Dialogue TC | | |
|---|---|---|---|---|---|---|---|---|---|
| | Macro-F1 | | | Acc. | | | Macro-F1 | | |
| completeness cases | whole | Complete word order | Incomplete word order | whole | Complete word order | Incomplete word order | whole | Complete word order | Incomplete word order |
| rate | 100.0 | 88.0 | 12.0 | 100.0 | 95.7 | 4.3 | 100.0 | 65.0 | 35.0 |
| baseline | | | | | | | | | |
| non-augmented | 81.6 | 81.3 | 83.5 | 79.7 | 80.7 | 56.9 | 64.6 | 64.9 | 64.4 |
| duplication | 82.3 | 82.9 | 78.0 | 82.4 | 83.6 | 56.5 | 68.4 | 70.2 | 67.5 |
| repetition | 83.5 | 84.7 | 74.6 | 81.4 | 82.3 | 63.3 | 69.2 | 70.3 | 68.6 |
| AEDA | 81.2 | 82.3 | 72.8 | 81.8 | 82.7 | 60.7 | 69.1 | 69.9 | 68.7 |
| EDA | 82.0 | 82.4 | 78.9 | 81.4 | 82.2 | 65.5 | 68.9 | 70.4 | 68.2 |
| ours | | | | | | | | | |
| $CM_{del}$ | 84.4 | 84.4 | 84.5 | 83.0 | 84.0 | 60.7 | 69.7 | 70.7 | 69.2 |
| $Shuf_{Sem.Presrv}$ | 84.2 | 83.9 | 86.0 | 81.8 | 82.4 | 68.1 | 70.2 | 71.1 | 69.7 |
| $Shuf_{Sem.Presrv}$ & $CM_{del}$ | 83.5 | 83.7 | 81.8 | 81.7 | 82.4 | 65.5 | 67.7 | 69.5 | 66.7 |
| $Shuf_{Sem.Non.Presrv}$ | 84.0 | 84.1 | 83.8 | 82.0 | 82.7 | 66.9 | 69.4 | 69.6 | 69.3 |
| $Shuf_{Sem.Non.Presrv}$ & $CM_{del}$ | 83.3 | 83.2 | 83.7 | 81.1 | 81.7 | 68.8 | 69.1 | 70.5 | 68.4 |

Table 11: Table analyzing the performance of models trained on augmented data in relation to word order completeness. The model augmented with $Shuf_{Sem.Presrv}$ demonstrates superior performance in test cases where word order is incomplete.

| | KLUE-TC | | | KLUE-NLI | | | Dialogue TC | | |
|---|---|---|---|---|---|---|---|---|---|
| | Macro-F1 | | | Acc. | | | Macro-F1 | | |
| completeness cases | whole | Complete Case Marker | Incomplete Case Marker | whole | Complete Case Marker | Incomplete Case Marker | whole | Complete Case Marker | Incomplete Case Marker |
| rate | 100.0 | 6.6 | 93.4 | 100.0 | 54.3 | 45.7 | 100.0 | 7.7 | 92.3 |
| baseline | | | | | | | | | |
| non-augmented | 81.6 | 94.0 | 80.7 | 79.7 | 79.7 | 79.6 | 64.6 | 68.4 | 64.3 |
| duplication | 82.3 | 94.0 | 81.5 | 82.4 | 84.8 | 79.6 | 68.4 | 71.5 | 68.2 |
| repetition | 83.5 | 96.0 | 82.6 | 81.4 | 83.0 | 79.6 | 69.2 | 71.7 | 69.0 |
| AEDA | 81.2 | 98.0 | 80.0 | 81.8 | 83.6 | 79.6 | 69.1 | 70.6 | 69.0 |
| EDA | 82.0 | 92.2 | 81.3 | 81.4 | 81.7 | 81.1 | 68.9 | 71.5 | 68.7 |
| ours | | | | | | | | | |
| $\text{CM}_{\text{del}}$ | 84.4 | 98.0 | 83.4 | 83.0 | 83.4 | 82.6 | 69.7 | 70.7 | 69.7 |
| $\text{Shuf}_{\text{Sem.Presrv}}$ | 84.2 | 98.0 | 83.2 | 81.8 | 82.1 | 81.3 | 70.2 | 72.3 | 70.0 |
| $\text{Shuf}_{\text{Sem.Presrv}}$ & $\text{CM}_{\text{del}}$ | 83.5 | 96.0 | 82.6 | 81.7 | 82.3 | 80.9 | 67.7 | 70.0 | 67.5 |
| $\text{Shuf}_{\text{Sem.Non.Presrv}}$ | 84.0 | 100.0 | 82.9 | 82.0 | 83.9 | 79.8 | 69.4 | 71.9 | 69.2 |
| $\text{Shuf}_{\text{Sem.Non.Presrv}}$ & $\text{CM}_{\text{del}}$ | 83.3 | 98.0 | 82.2 | 81.1 | 81.1 | 81.1 | 69.1 | 70.5 | 69.0 |

Table 12: Table analyzing the performance of models trained on augmented data in relation to case marker completeness. The model augmented with $\text{CM}_{\text{del}}$ demonstrates superior performance in test cases where case markers are incomplete.

| python Ver. | 3.8.0 |
|---|---|
| pyTorch Ver. | 2.0.0+cu117 |
| transformer Ver. | 4.30.1 |
| epoch | 3 |
| batch size | 16 |
| learning rate | 5.00e-05 |

Table 13: Development environments

```python
messagesHeader = [
    {"role": "system", "content": "You are a helpful assistant"}
]

def inputMaker(sentence):
    ret = []

    prompt = '''
다음 문장을 의미가 변하지 않게 어순을 변경해줘.
문장의 각 단어는 그대로 유지되어야해
답은 단답형으로 해줘

입력 문장: {}

변경된 문장은?:
'''.format(sentence)
    ret.append({"role": "user", "content": "{}".format(prompt)})
    return ret

def call_charGPT(turn) :
    response = openai.ChatCompletion.create(
        engine="deply-ranker-gpt35turbo",
        messages=messagesHeader + turn #+ messageTail
    )
    return response
```

Figure 2: ChatGPT prompt and version information

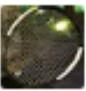 다음 문장을 의미가 변하지 않게 어순을 변경해줘.
문장의 각 단어는 그대로 유지되어야 해

입력: 나비가 꿀을 마신다.

변경된 문장은?

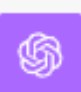 변경된 문장: 꿀을 나비가 마신다.

Figure 3: Example of chatGPT prompt usage

