# OpenReview forum: "Does Incomplete Syntax Influence Korean Language Model? Focusing on Word Order and Case Markers"
_colmweb.org/COLM/2024/Conference — COLM_

### Official Review · Reviewer_tuTs · 2024-04-28

**Rating:** 8
**Confidence:** 4
**Ethics Flag:** 1

**Summary:**

This paper introduces a Korean data set to test out how syntactic variation found in Korean speech and certain common writing styles (dropping of case markers and non-SOV word order) affects LLM tasks.  In addition to the data set itself, the authors bench mark on 3 Korean LLMs across 2 datasets and 4 tasks, looking at fine-tuned and non-fine tuned models.

Although such syntactic variation is common, as the authors show as part of the dataset construction, the LLMs perform much better when fine-tuned with such data than when used out of the box.

As the authors point out, in addition to providing a dataset for Korean, this type of syntactic variation is common in many languages and so the findings and the approach to creating the dataset are applicable well beyond Korean.

**Questions To Authors:**

Please proofread the final paper, paying special attention to spacing around parentheses and to determiner placement. Also, references to "chapter" should say "section".

Comments in order of the paper:

Make clear the license of the dataset being made available and whether any code will be made available with it (e.g. code for its creation in case others want to replicate this for another language or evaluation code).

The overall description of Korean may need to be a bit clearer for those who know nothing about Korean syntax, which will be most of the readers.  Early on make clear (perhaps near where Fig 1 is discussed and repeated in Sec 3) that:
- Korean is by default SOV
- The main reorderings are among full phrases (e.g. NP arguments) and occur before the verb
- Moving the verb earlier is much rarer
- Splitting apart an NP argument (e.g. fronting an adjectival modifier but not the noun) is much rarer
- Natural dropping of case markers is more common with subjects than objects (or vice versa or the same, whichever is true)

For all the tables of results, indicate statistical significance, or lack thereof, between the best system and the next best and between the best system and the baseline.

p1: It is not clear what is meant by "different meanings".  Is this argument structure different (e.g. if the original Korean was "dog cat bit" and the new order was "cat dog bit" both with no case markers, then the subject and object are swapped) or discourse structure different (e.g. the first NP will be interpreted as a topic).

p2: For the conversational corpus, have the participants all agreed to have their data used? This is about the existing corpus, but it would be good to make clear that all ethical considerations were met in creating the corpora.  To a lesser extent, for the non-conversational corpus, are all copyright issues respected?

Table 1: It would be interesting to know the case marker drop rate for subjects vs objects.  This would be easy to calculate on the 300 sentences and could be added as a footnote.
It would be good to know the % of sentences which have both non-canonical word order and case marker dropping (as opposed to one or the other).
What is "Including Restore"?

Sec 3: when calculating the correction rate, were the words counted or the constituents (e.g. NPs)?  Although more of a hassle to compute, the constituent rate would be more indicative since the paper is really about word order changes of entire constituents such that a single word constituent ("dog") counts the same as a complex on ("large brown dog with spots").

Table 2: what is "rshuf"? Presumably the non-V-final orders are much rarer; this should be mentioned earlier. Also, the example has the object still with the verb, which maintains the VP (if Korean has one, which I realize is a major discussion), was the V-subj-obj order also used?

Sec 4: Make clear that created this much data to allow fine-tuning. 22K sentences are not needed for eval.

Table 3: In the caption mention that the metrics are "Macro F1", "Accuracy" and "ROUGE"  (I think ROUGE is spelled with all caps, but please check).

sec 5.3: Mention that you also tried the LMs out of the box without fine-tuning.

sec 6.2: Was this 5-fold cross-validation or simply presenting the data in different orders to see the variants. State briefly (due to space) how much variance there was.

Sec 7: Although not required, if you have references for the word order and case marking facts for these languages, include them.  This is, admittedly, all well know in the linguistic community, but less well known to many NLP practitioners.

Table 9: Show the Korean version as well. Perhaps merge this with Fig 3.

App D: EDA does require a way to get synonyms.  Most languages have this and it is even easier with LLM supported languages, but it does add complexity, especially if the substitution requires inflecting the form.

**Reasons To Accept:**

- Highlights major issue between standard language and language in actual usage and how this affects LLM-based tasks.
- Quantifies the issue.
- Provides straight-forward way to create a data set to fine-tune existing LLMs.
- Provides the data set.
- Bench-marks on 3 models, 2 datasets, and 2 tasks.
- Findings and solution are very broadly applicable outside of English.

**Reasons To Reject:**

- Minor rewriting needed to make the Korean facts clearer to those without linguistic knowledge of Korean.
- Needs proofreading.

---

> ### Author Rebuttal · Authors · 2024-05-30
>
> Firstly, we sincerely thank you for your deep understanding and interest in our research. Due to the rebuttal length limitation (2500 characters), we apologize for not being able to explain each point in detail.
>
> 1. Knowledge of Korean Grammar:
> The information you shared is essential for understanding our research. We will include this in the revised version.
>
> 2. Licenses and Usage Permissions:
> All data used is anonymized and publicly available for research. Dialogue data cannot be redistributed in modified form, so only non-dialogue data from the SIKO dataset will be made public. We will re-examine all licenses and permissions.
>
> 3. Additional Information:
> We will ensure that the revision includes details on the usage of SIKO data, table metrics, cases without tuning, references for Korean grammar, the thesaurus used in EDA, and the Korean version of the prompts used, as you pointed out.
>
> 4. Responses to Specific Questions:
> Many questions arose due to our insufficient explanations. We will clarify these points in the revision.
>
> "Different meanings": Refers to potential misinterpretation of the author's intent.
>
> Omitted case markers: Investigating the role of omitted case markers is indeed interesting. We will include this in the revision as adding new research results during the review period is prohibited.
>
> "Including Restore": We restored all omitted case markers in 300 samples and then investigated the rate of omitted case markers. "Including Restore" indicates that the number of case marker positions includes the restored case markers.
>
> Correction rate unit: Based on words, evaluated by human evaluators with simple and specific guidelines for consistency.
>
> Meaning of "rshuf": Refers to "randomly shuffled data" mentioned earlier. We missed this detail. Korean word order is relatively free, but the verb position tends to stay fixed. Our changes include cases where the verb is not in the final position, especially in random shuffles and meaning-preserving word order changes (e.g., chat conversations).
>
> Verification method: To avoid sampling bias, we performed five different samplings and expansions for each expansion rate, conducted tuning, and measured performance. We will investigate data variation.
>
> Lastly, we deeply appreciate your valuable feedback, which has improved our paper. Your advice included many points we had not considered, providing valuable information for other researchers. We will ensure all suggestions are reflected in the revision.

---

### Official Review · Reviewer_8Me2 · 2024-05-08

**Rating:** 6
**Confidence:** 4
**Ethics Flag:** 1

**Summary:**

The paper analyzes the effect of incomplete syntactical structures in Korean sentences for LLMs. The paper focuses on two forms of incomplete syntax, which are omitting case markers and changing word order. In addition, a new syntactically incomplete Korean dataset (SIKO) was compiled and this dataset is also used for fine-tuning the LLMs. Two Korean LLMs were used. The experiments showed that LLMs are flexible in handling the incomplete syntactic structures.

**Questions To Authors:**

- Please use "section" instead of "chapter".
- Section 4.1: Please specify the accuracy of the morphological analyzer used and also state how much error occurred in the process of stripping out the case markers.
- "Mixed Application(" -> "Mixed Application ("
- Page 6: What is ICL?
- Section 5.4: "multiple human-centric tasks such as evaluation, restoration, and correction were conducted.". What is the reason of these tasks in the paper?
- "Two workers". Not a good phrase. May be "Two people" or something like.
- "tables 5 and 6" -> "Tables 5 and 6"

**Reasons To Accept:**

- The paper analyzes the effect of syntactically incomplete sentences on LLM, which can be regarded as one of the first studies in this area.
- The paper compiles a new syntactically incomplete Korean dataset.

**Reasons To Reject:**

- The paper conducts an experimental analysis, but there is no novelty in methodogical aspects of the models used.
- Some explanations in the paper are not clear and there are some problems with the language use.

---

> ### Author Rebuttal · Authors · 2024-05-30
>
> First, we would like to express our gratitude for your deep understanding and interest in our research.
>
> We appreciate your point that the deletion of case markers and changes in word order may not be methodologically novel, and we agree to a certain extent. However, we wanted to share with other researchers that even such familiar and simple methods can contribute to performance improvement. We will prepare a comprehensive rewriting to make our research more easily understandable to other researchers.
>
> Response to Questions:
>
> Thank you for pointing out the deficiencies related to formatting. We believe these are crucial elements for enhancing the clarity of our research. We will definitely incorporate the feedback you provided.
>
> The morphological analyzer we used was developed in-house for the services of the company I am affiliated with. In terms of general performance, it has recorded an F1 score of 98.826 in evaluations conducted on sampled news article sentences.
>
> It seems we did not provide sufficient explanation when using the term "ICL." This term was used to indicate that the task was performed in a zero-shot manner with no contextual information provided. (In-context learning)
>
> The human evaluators you mentioned were involved in the tasks described in Section 3 (Table 1) and Section 4.2. In Section 3, they identified the presence of omitted case markers or altered word order in general Korean usage and performed the task of restoring omitted case markers or word order to calculate their scale. In Section 4.2, they reviewed and corrected the meaning-preserved word order change data generated by ChatGPT to ensure that the meaning was indeed preserved.
>
> Once again, thank you for your understanding and valuable advice.

---

> > ### Comment · Reviewer_8Me2 · 2024-06-03
> >
> > I thank the authors for the clarifications.

---

> > > ### Author Response · Authors · 2024-06-06
> > >
> > > We deeply appreciate your profound understanding and encouragement.

---

### Official Review · Reviewer_51wH · 2024-05-10

**Rating:** 6
**Confidence:** 4
**Ethics Flag:** 1

**Summary:**

- This paper investigates the performance of LLMs for incomplete syntactic Korean dataset.
- This paper finds that Korean LLM exhibits flexibility when dealing with syntactically incomplete inputs.

**Reasons To Accept:**

- The findings in this paper could be generalized to other languages that has similar characteristics, such as Japanese.

**Reasons To Reject:**

- Lack of experiments. Specifically, more experimental results with other languages make clearer the generalizability of this paper.

---

> ### Author Rebuttal · Authors · 2024-05-30
>
> First, we would like to express our gratitude for your deep understanding and interest in our research.
>
> As you mentioned, conducting experiments on various languages would have increased the generalizability and value of our research. We have also investigated languages with similar features, such as Japanese (which uses case markers), and languages with relatively free word order like Turkish, Magar, and Kazakh.
>
> However, we believed that grammatical knowledge was necessary for preparing and analyzing the experiments, so we limited the scope of our research to our native language, Korean. If given the opportunity, we plan to expand the application of syntactically incomplete data to more languages in future research.
>
> Once again, thank you for your understanding and valuable advice.

---

> > ### Comment · Reviewer_51wH · 2024-06-05
> >
> > Thank you for your comment.
> >
> > I agree that the experiment with a single language, Korean, is suffice as a first-step research paper.
> > I hope that you and followers of this paper will add experimental results through the experiments with other languages in the future.

---

> > > ### Author Response · Authors · 2024-06-06
> > >
> > > Dear Reviewer,
> > >
> > > We deeply appreciate your valuable feedback and suggestions, which have provided us with an opportunity to thoroughly review and improve our research. Your encouragement has also been a great source of strength for us.
> > >
> > > Thank you very much.

---

### Decision · Program_Chairs · 2024-07-10

**Decision:**

Accept

**Comment:**

This paper introduced a benchmark in Korean that focuses on syntactic incompleteness in the language; finetuning on this dataset greatly improves the performance of LLMs highlighting the importance of syntax in Korean. Overall, reviewers liked this paper but had some concerns on writing quality and provided many constructive suggestions to help make the paper better. The paper contributes a useful dataset and provides a concrete improvement to language technologies beyond English.

[At least one review was discounted during the decision process due to quality]